# Field-deployable branch enclosure system for biogenic volatile organic compounds emitted from conifers

4

- Yuki Ota<sup>1,2</sup>, Takuya Saito<sup>1,2</sup>, Stephen J. Andrews<sup>3</sup>, Tetsuo I. Kohyama<sup>1</sup>, Yoshihisa Suyama<sup>4</sup>, Yoshihiko Tsumura<sup>5</sup>,
- Tsutom Hiura<sup>1</sup>
- <sup>1</sup> Department of Ecosystem Studies, Graduate School of Agricultural and Life Sciences, The University of Tokyo,
- Tokyo, 113-8657, Japan
- <sup>2</sup> National Institute for Environmental Studies, Tsukuba, 305-8506, Japan
- 8 9 <sup>3</sup> Wolfson Atmospheric Chemistry Laboratories, Department of Chemistry, University of York, York, YO10 5DD,
- 10 11
  - <sup>4</sup> Field Science Centre, Graduate School of Agricultural Science, Tohoku University, Osaki 989-6711, Japan
- 12 <sup>5</sup> Faculty of Life & Environmental Sciences, University of Tsukuba, Tsukuba, 305-8572, Japan
- 13 Correspondence to: Yuki Ota (yukiota0007@gmail.com) and Takuya Saito (saito.takuya@nies.go.jp)

14

28

- Abstract. Biogenic volatile organic compounds (BVOCs), emitted primarily from terrestrial plants, significantly
- influence atmospheric chemistry and climate change. Conifers are major sources of BVOCs in temperate regions.
- However, their unique physiology, particularly the storage of terpenes within their tissues, makes accurate
- measurements of BVOC emissions challenging owing to contact-induced release. We developed a portable
- dynamic branch enclosure system specifically optimized for BVOC measurement in conifers, which enables
- measurements of multiple trees in a single day. The system uses filtered ambient air as a purge gas to reduce
- logistical challenges and features a foldable bag design to minimize excessive BVOC release induced by physical
- contact. It provides BVOC- and ozone-free air, maintains stable internal temperature and humidity conditions
- closely approximating those of natural environments, and ensures repeatable measurements of BVOC emissions.
- Field testing with Japanese cedar (Cryptomeria japonica) demonstrated the system's robust field performance,
- successfully capturing both significant inter-individual variability and the dynamic diurnal patterns of BVOC
- emissions. The system's ability to reliably resolve these differences under field conditions demonstrates its
- applicability for advancing our understanding of BVOC dynamics in diverse ecosystems.

## 1. Introduction

- Biogenic volatile organic compounds (BVOCs), emitted primarily from terrestrial plants, play a significant role in
- atmospheric chemistry. Through atmospheric degradation processes, BVOCs contribute to the formation of
- tropospheric ozone and secondary organic aerosols (Chatani et al., 2015; Ghirardo et al., 2016; Laothawornkitkul
- et al., 2009), which significantly affect air quality. These atmospheric constituents can alter radiative forcing and
- precipitation patterns, influencing the global climate (Randall et al., 2013; Rotstayn et al., 2009).
- Among terrestrial plant species, conifers are major BVOC emitters in temperate regions, and their emissions have
- important implications for regional air quality and climate (Peñuelas and Staudt, 2010). They release a diverse
- array of BVOCs, notably monoterpenes (MTs), sesquiterpenes (SQTs), and diterpenes (DTs) (Chatani et al., 2018;
- Guenther et al., 1993, 2012; Matsunaga et al., 2011). Monoterpenes (C<sub>10</sub> compounds) are highly volatile and major
- BVOCs in temperate and boreal regions (Guenther et al., 1993). Sesquiterpenes (C<sub>15</sub>) have a notable role in
- atmospheric processes, particularly in particle formation, as their oxidation produces ultra-low volatility organic

compounds that act as efficient nucleators (Dada et al., 2023). Diterpenes ( $C_{20}$ ) have even lower volatility but high reactivity, which may make them particularly important for particle formation, although these properties also pose challenges for their observation.

The profiles of BVOC emissions from plants are determined by a complex interplay of factors, including environmental conditions such as temperature and drought (Birami et al., 2021), genetic factors, and biotic stresses such as plant–microbe interactions (Saunier et al., 2020). Moreover, some plant species alter their emission profiles in response to BVOCs emitted by neighbouring plants (Arimura et al., 2012). However, how these factors influence the composition and rates of BVOC emission remains insufficiently understood (Tani et al., 2024).

Recent comprehensive reviews have further underscored the critical and distinct roles of these terpene classes in biosphere-atmosphere interactions (Bourtsoukidis et al., 2024, 2025; Yañez-Serrano et al., 2024). For instance, diterpenes are now understood to be particularly potent contributors to SOA formation, potentially having a disproportionately large impact relative to their emission rates (Yañez-Serrano et al., 2024). Moreover, these reviews highlight that the emission rates and composition of MTs and SQTs can vary significantly among individuals, which may reflect diverse adaptive strategies to environmental stresses (Bourtsoukidis et al., 2024, 2025). To untangle the complex factors governing these emissions, a dual approach of broad-scale analysis and detailed, individual-level data collection is essential. Quantifying emission characteristics at the level of individual plants is crucial for understanding how emissions change in response to varying environmental conditions. This understanding is essential for predicting future changes in BVOC–aerosol–climate feedbacks.

One challenge hindering individual-level BVOC measurements under natural conditions is the logistical difficulty associated with conventional sampling methods. Since conifers store terpenes within their needles (Saito et al., 2022), physical contact can trigger a burst release of these compounds, making it difficult to accurately measure their emission rates. The cuvette method, which clamps an individual leaf, is generally unsuitable for conifers for this reason. The dynamic branch enclosure method (Ortega et al., 2008; Ortega and Helmig, 2008) reduces direct leaf contact but still contacts the stem during device installation, which is particularly important consideration for conifers such as cedars that store terpenes in both leaves and stems. Several studies using dynamic branch enclosure methods on coniferous trees allowed a stabilization period after device installation of approximately 24 h (Helin et al., 2020; Helmig et al., 2013; Hiura et al., 2021; Matsunaga et al., 2011, 2012, 2013), 48 h (Bouvier-Brown et al., 2009), or 1-2 weeks (Praplan et al., 2020) to mitigate the effects of excessive emissions associated with physical contact during device installation. During the stabilization period, it is essential to either maintain continuous gas flow (Helin et al., 2020; Helmig et al., 2013; Hiura et al., 2021; Matsunaga et al., 2011, 2012, 2013) or avoid fully sealing an enclosure (Praplan et al., 2020) to sustain normal physiological conditions and prevent condensation before measurements are taken. However, when gas cylinders are used as the purge gas source in locations without a power supply, numerous cylinders must be transported to the site to maintain continuous gas flow throughout the measurement period, posing significant logistical challenges for field measurements (Hiura et al., 2021; Matsunaga et al., 2011, 2012, 2013). Instead of gas cylinders, pumps can be used to supply purge air by drawing in external air. However, to our knowledge, no studies have clearly shown whether a pump-based system can be entirely powered by a portable battery, which is essential in remote areas without a power supply. These difficulties highlight the need for simplified, field-deployable techniques for BVOC measurement.

Here, we developed a simplified dynamic branch enclosure system for measuring BVOC emissions from conifers, which enables measurements of multiple trees in a single day. Our system addresses the limitations of conventional

methods by using filtered ambient air as the enclosure purge gas and minimizing purge air consumption with a foldable bag design. The introduction of a foldable bag eliminates the need for purge gas during equipment setup. It enables a single person to measure BVOCs from up to five trees per day using the dynamic branch enclosure method. We describe the design and evaluation of our enclosure system and demonstrate its applicability through measurements of BVOC emissions from Japanese cedar (*Cryptomeria japonica*), one of the dominant conifer species in Japanese forests.

#### 2. Methodology

#### 2.1 Branch enclosure system

# 2.1.1 System design and components

The branch enclosure system is designed to deliver VOC- and ozone-free air while ensuring stable internal conditions for BVOC measurements (Fig. 1a). It uses a diaphragm pump (FD-15, IBS, Osaka, Japan) and an activated carbon cartridge (CHC-50, Advantec, Tokyo, Japan) to purify the purge air. The chamber consists of an open-ended fluorinated ethylene propylene (FEP) bag (25 cm width × 60 cm long, GL Sciences Inc., Tokyo, Japan) to enclose the branch under observation. To ensure circulation and homogenization of air inside the bag, purified air is introduced through 6 mm PTFE tubing formed into 2 rings, each containing 12 holes (approximately 1.5 mm in diameter) for the air to exit from. The 2 rings also provide structural integrity to avoid contact with the branch. The PTFE rings are supported by a tripod equipped with a flexible arm to minimize stress on the branch (Fig. 1b). A mass flow controller (D-6361-DR/FAS, Bronkhorst Japan, Tokyo, Japan) maintains a constant purge air flow rate of 5 L/min. Power for the mass flow controller and the diaphragm pump is supplied by a portable battery (JE-1000D, 30.4Ah/35.2V DC, Jackery, Tokyo, Japan). The temperature inside the bag and the nearby photosynthetically active radiation (PAR) are monitored by a PTFE-coated thermocouple (GL Sciences Inc., Tokyo, Japan) and a PAR sensor (MIJ-14PAR Type 2/K2, Environmental Measurement Japan, Fukuoka, Japan), respectively, using a data logger (Thermic model 2400A, ETOdenki, Tokyo, Japan). All purge lines were made with 6 mm PTFE tubing. The enclosure bag and PAR sensor were secured at an appropriate height on a tripod. A photographic white umbrella was used to diffuse and reduce the intensity of direct sunlight on the enclosure during purging and sampling.

Figure 1: (a) Schematic of Branch enclosure and associated apparatus, (b) enclosure setup, and (c) close-up of the enclosure. MFC, mass flow controller; Logger, data logger; PAR, photosynthetically active radiation sensor.

# 2.1.2 System operation and sampling protocol

The system is designed for efficiently sampling multiple trees in a single day. This is achieved by pre-installing the support collars on each target tree the day before, and then moving the main portable enclosure apparatus between these collars on the sampling day. In this study, five sets of collars were used to sample five trees. The detailed procedure for enclosing a single branch is as follows:

- 1. Pre-attachment: One day before sampling, an open-ended FEP bag and a funnel-shaped support, referred to as an 'Elizabethan collar', are attached to a branch (Fig. 1a). The collar is used to minimize contact between the bag and the branch. The trunk end of the FEP bag is secured around the branch with hook-and-loop tape. The bag is folded back so that it remains off the collar.
- 2. Enclosure deployment: On the sampling day, the PTFE ring is inserted over the branch, the branch is enclosed

- by unfolding the FEP bag, and the distal end of the bag is secured around the branch with hook-and-loop tape.

  After the branch is enclosed, the inside of the bag is purged with the purified air for at least 1 h before BVOC collection. The bag is secured to maintain a slight positive pressure inside, ensuring that purge air is discharged from both ends.
- BVOC collection: Air in the bag is drawn through an adsorbent tube (6 mm OD × 90-mm-long glass tube; Supelco, Bellefonte, PA, USA) at 200 mL/min for 10 min by a portable sampler (GSP-300FT-2, Gastec, Kanagawa, Japan). Two types of sorbent tubes are used: one for MTs, packed in series with Tenax TA (20/35 mesh, approximately 100 mg; Supelco) and Carbopack B adsorbents (20/40 mesh, approximately 50 mg; Supelco), and another for SQTs and DTs, packed with HayeSep Q adsorbent (60/80 mesh, Hayes Separations Inc., Bandera, TX, USA). After sampling, compounds collected on the HayeSep Q adsorbent tubes are immediately extracted with approximately 2 mL of hexane (special grade; Fuji Film Wako, Osaka, Japan). The solvent is transferred into 2 mL glass vials, stored in a cooler box (approximately 0 °C) for 1 day, and then frozen (approximately below -30 °C) until analysis. The MT sorbent tubes are also stored in a cooler box and then frozen until analysis.

#### 2.2 Measurements

#### 2.2.1 Monoterpenes

The MT adsorbent tubes were analysed by a custom-built thermal desorption unit coupled to a gas chromatograph (GC) equipped with a mass selective detector (MSD) and a flame ionization detector (FID) (Agilent 6890/5973, Agilent Technologies, Santa Clara, CA, USA). The adsorbent tubes were purged at 40 °C with nitrogen at 40 mL/min for 1 min, then heated to 250 °C. The desorbed analytes were transferred in helium at 10 mL/min to a focusing trap packed with 2 mg Tenax TA and 2 mg Carboxen 1000 (Supelco) maintained at –130 °C. The trap was then heated to 180 °C, and desorbed analytes were injected onto an HP-5 column (60 m, 0.32 mm I.D., 1 µm film thickness; Agilent) in the GC oven. The GC oven temperature was held at 35 °C for 2 min, increased to 160 °C at 4 °C/min, then to 300 °C at 45 °C/min, and held at 300 °C for 10 min. Analytes eluting from the column were split between the MSD and FID. Methyl salicylate was also analyzed using this method. MSD analysis was performed in SIM/SCAN mode. Peaks were identified by comparing retention times with those of standards (Table S1) prepared in methanol. MTs were quantified using the FID, based on calibration curves constructed from the peak areas of standards introduced at various concentrations.

## 2.2.2 Sesquiterpenes and diterpenes

Following solvent extraction and concentration (Matsunaga et al., 2012), SQTs and DTs were measured by GC-MSD/FID (Agilent 6890/5973). An internal standard (approximately 10 ng of cyclopentadecane dissolved in hexane at a concentration of around 10 ng/μL; Tokyo Chemical Industry Co., Ltd., Tokyo, Japan) was added to the hexane extract, which was then concentrated to approximately 30 μL by using nitrogen blowdown at 60 mL/min. A 1μL aliquot of the concentrated extract was manually injected into the GC. The GC oven temperature was programmed to hold at 60 °C for 2 min, ramp to 120 °C at 30 °C/min, increased to 150 °C at 2 °C/min, and finally ramped to 320 °C at 5 °C/min, holding at 320 °C for 10 min. The separated compounds were split between MSD and FID, with MSD operated in SIM/SCAN mode; quantification was performed using the SIM mode signals.

Peaks were identified by comparing retention times with those of standards (Table S1). SQTs and DTs were quantified by using calibration curves based on the peak areas of standards introduced at various concentrations.

#### 2.3 Calculation of basal emission rate

The rate of BVOC emission  $(E, \text{ in ng } (\text{gdw})^{-1} \text{ h}^{-1})$  was first calculated using a mass-balance equation:

$$E = \frac{F \times (C_{\text{out}} - C_{\text{in}})}{W_{\text{dry}}} \tag{1}$$

- where F is the flow rate of purge air through the enclosure (L h<sup>-1</sup>);  $C_{\text{out}}$  is the BVOC concentration in the air exiting the enclosure (ng L<sup>-1</sup>), determined from the mass of the compound collected on a sorbent tube divided by the total volume of air sampled;  $C_{\text{in}}$  is concentration of BVOCs in the incoming purge air, determined from a blank measurement (an empty enclosure);  $W_{\text{dry}}$  is the dry weight of the enclosed branch (g dw), estimated from Shirota (2000).
- To allow for comparison across measurements taken at different temperatures, this measured emission rate (E) was then normalized to a basal emission rate ( $E_s$ , in ng (gdw)<sup>-1</sup> h<sup>-1</sup>) at a standard temperature ( $T_s$ , 30°C = 303.15 K), following the algorithm of Guenther et al. (1993):

$$E_s = \frac{E}{\exp[\beta(T - T_s)]} \tag{2}$$

where T is the temperature inside the enclosure, and  $\beta$  is an empirical coefficient that quantifies the temperature sensitivity of emissions. The  $\beta$  values used were 0.17 for MTs, 0.20 for SQTs, and 0.21 for DTs (Matsunaga et al., 2011, 2012, 2013). It should be noted that these  $\beta$  values were not empirically derived from our own dataset, as the measurements were conducted over a narrow temperature range that was unsuitable for robust parameterization.

## 3. Method evaluation

## 3.1 Performance of activated carbon cartridge

- We evaluated the efficiency of the activated carbon cartridge at removing BVOCs and ozone. Cedar branches collected from trees growing in the premises of the National Institute for Environmental Studies were chopped into pieces enclosed in a FEP bag. BVOC emission rates were measured with and without the activated carbon cartridge (n = 4–6). Ozone removal efficiency was similarly evaluated by measuring ambient ozone with and without the cartridge using a UV absorption ozone monitor (1006-AHJ, Dasibi, Glendale, CA, USA). Ozone was measured for 10 min a day over 2 days.
- Total BVOC concentration was significantly lower with the cartridge than without (P=0.0117, Welch's two-sample t-test; Table 1). Ozone concentrations in the air passing through the activated carbon cartridge were significantly lower than ambient (Table 1). This confirms that the activated carbon cartridge effectively removes ozone, consistent with findings from a previous study (Namdari et al., 2021). These findings confirm that our system is capable of supplying ozone- and BVOC-free air to the enclosure.

Table 1. Ozone mixing ratio and total concentration of the measured BVOCs with and without the activated carbon cartridge. Both new and used activated carbon cartridges were tested. The used cartridge had been employed for

approximately 8 hours in field experiments. Results obtained with the used cartridges are shown in parentheses. The limit of detection (LOD) was 1 ppb for ozone and 0.77 ng  $L^{-1}$  for total BVOCs. The LOD for total BVOCs was calculated by taking the root sum of squares (RSS) of the individual compound LODs (Table S1) and dividing by the 2 L sampling volume.

|                            | Ozone (ppb) | Ozone (ppb) Total BVOCs (ng L <sup>-1</sup> )                              |  |                              |
|----------------------------|-------------|----------------------------------------------------------------------------|--|------------------------------|
| No cartridge               |             | 21 ± 1                                                                     |  | $2400 \pm 1800$              |
| Activated carbon cartridge |             | <lod (<lod)<="" td=""><td></td><td><lod (<lod)<="" td=""></lod></td></lod> |  | <lod (<lod)<="" td=""></lod> |

#### 3.2 Environmental control

We evaluated the effect of purge gas source on environmental conditions inside the bag enclosure, using purge gas from ambient air (as described in section 2.1.2) and from gas cylinders (compressed air, Tomoe Shokai Co., Ltd., Tokyo, Japan) as used in previous studies (Hiura et al., 2021; Matsunaga et al., 2011). The experiments were conducted at Tsukuba University Mountain Science Centre. During the experiments, the temperature and relative humidity inside the bag were measured every minute with a hygrometer (2119A, ETO denki, Tokyo, Japan) and a PTFE-coated thermocouple (GL Sciences Inc., Tokyo, Japan). Those outside the bag were obtained from a meteorological station at Tateno, Tsukuba, operated by the Japan Meteorological Agency.

Our system is designed to maintain internal conditions that closely reflect the ambient environment. The humidity within the enclosure more closely reflected the external humidity when purge air was supplied via our air delivery system compared to using dry cylinder gas (Table 2). This reduces the atmospheric water demand on the branch.

within the enclosure more closely reflected the external humidity when purge air was supplied via our air delivery system compared to using dry cylinder gas (Table 2). This reduces the atmospheric water demand on the branch, avoiding potential artifacts caused by an artificially high vapor pressure deficit (Stoy et al., 2021; Núñez et al., 2002). The temperature within the enclosure remained consistent with the external temperature, regardless of whether purge air was supplied via the air delivery system or from a cylinder (Table 2). In some cases, Praplan et al. (2020) found that prolonged direct sunlight causes an enclosure to heat up, resulting in a maximum temperature difference of 27.5°C. In this study, it is assumed that the use of a sunshade prevented direct sunlight, thereby mitigating the temperature increase. This overall stability in both temperature and humidity is attributed to two key design features. First, the photographic white umbrella was used to diffuse and reduce the intensity of direct sunlight, preventing overheating of the enclosed branch. Second, the high air exchange rate, with a residence time of approximately 1.6 minutes, effectively prevents the accumulation of both heat and transpired water vapor.

Table 2. Temperature and relative humidity inside and outside the bag enclosure when air supply was drawn from cylinder and ambient air

|                  | Air source       |                    |                  |                   |  |
|------------------|------------------|--------------------|------------------|-------------------|--|
|                  | Cylinder         |                    | Ambient air      |                   |  |
|                  | Inside enclosure | Outside enclosure  | Inside enclosure | Outside enclosure |  |
| Humidity (%)     | 40 ±             | $1 	 70 \pm 3$     | $59 \pm 2$       | $63 \pm 0.9$      |  |
| Temperature (°C) | $30.4 \pm 0.$    | $6 	 29.0 \pm 0.4$ | $30.0 \pm 1$     | $27.7 \pm 0.4$    |  |

# 3.3 Measurement repeatability

We evaluated the repeatability of BVOC emission rate measurements using a 1-m-long branch collected from a 10-year-old cedar tree at the Tsukuba University Mountain Science Centre. After trimming the base, the branch was re-cut under water to maintain its vascular integrity, and the cut end was kept submerged throughout the experiment. This underwater cutting technique is a standard method to prevent air from the xylem vessels, which can cause cavitation and disrupt water transport (Ogasa et al., 2016; Umebayashi et al., 2016). Indeed, measurements on detached branches represent a well-established approach in BVOC research (e.g., Jardine et al., 2020), including for coniferous species with large storage pools similar to *C. japonica* (Mochizuki et al., 2011; Miyama et al., 2018). Furthermore, Monson et al. (2007) demonstrated that this method maintains stable rates of photosynthesis, stomatal conductance, and isoprene emission for detached branches, showing no significant differences from branches that remained attached to the tree. As *C. japonica* emits stored rather than de novo synthesized BVOCs, and the distance from the cut site to the enclosed section of the branch was sufficiently long (60 cm), the effect of cutting on our measurements is considered negligible. Following the procedure described in section 2.1.2, air in the enclosure bag was collected in adsorbent tubes (9 tubes for MTs, 5 tubes each for SQTs and DTs). Blank samples were also taken from the empty enclosure.

The relative standard deviations (RSDs) of MTs ranged from 4.1% to 19%, with an average of 10%. That of SQT was 5.6% (Fig. 2). That of individual-level BVOCs in the field was 159% (data not shown).

Figure 2: Basal emission rates of terpenes ( $E_s$ ) and relative standard deviations (%, n = 5, blue)

These findings suggest that RSDs of the new measurement method are sufficiently low compared with the variability in individual-level BVOC emission rates, confirming the method's reproducibility in assessing individual emission rates. The RSDs are comparable to those reported in previous studies with other tree species (Li et al., 2019).

In addition to the tests on excised branches, we conducted a reproducibility experiment on an intact branch of a live seedling. The same branch was measured twice, and the results are compared in Figure S1. The measurements showed reasonable consistency for all detected compounds, supporting the stability of our system. We note, however, that this test was conducted on a young seedling with a limited number of emitted compounds.

#### 3.4 System stabilization time

We conducted an indoor experiment to evaluate the time required for the excessive release of BVOCs due to contact during the attachment of the enclosure system to subside. A cut cedar branch placed in a thermostatic chamber was used. Measurements were taken with a TDU-GC-MS system, as in section 2.2.1, but equipped with a dual trapping system for online preconcentration and refocusing of BVOCs. Air was drawn from the enclosure at 45 mL/min into the TDU for 12 min and analysed by GC-MS. Measurements were taken every 45 min over 3 days. Throughout the experiment, the chamber was maintained at a constant 25 °C. As the temperature remained constant, no temperature correction was applied in the calculation of the basal emission rate.

We also investigated effects of extending the pre-attached FEP bag on basal emission rates during the measurement

day. Using the same materials as before, we tied a FEP bag to a cedar branch, folded it, and left it for approximately 24 h. The bag was then extended to cover the branch again. Samples were taken every 45 min, 20 times in total, while the bag was purged with air, and we compared basal emission rates before and after the bag was extended. Installation of an enclosure (i, Fig. 3) initially triggered an excessive release of BVOCs. Emission decreased to approximately 5% of its peak value within 24 h of installation.

Figure 3. BVOC emission rates from a tree branch over time under different enclosure conditions. A branch was covered with a bag (0 h, i), and continuous measurements were taken until 45 h. The bag was folded back (ii), and this state was maintained until 52 h (iii). The bag was unfolded over again to cover the branch, and measurements were resumed until 72 h. Total amount of terpenes emitted was 3.75 μg/gdw. Considering that the amount of terpene stored in the leaves of Cryptomeria is approximately 15.877mg/gdw (Saito et al., 2022), this represents only 0.02% of its storage pool.

These results indicate that allowing 24 h after enclosure installation mitigates potential overestimation of BVOC emission rates due to the installation. Previous studies allowed 24 h or more (Helin et al., 2020; Helmig et al., 2013; Hiura et al., 2021; Matsunaga et al., 2011). Our results suggest that 24 h is enough.

The293 pre-294 suff

The results also reveal that the emission rate 1 h after bag unfolding (Fig. 3, iii) was virtually unchanged from the pre-unfolding level (ii), so the potential effect of excessive emissions during the bag-unfolding process was sufficiently accounted for. Notably, Mochizuki et al. (2011) showed that the excessive emission of MTs induced

by vibration stimulus in Japanese cedar subsided within approximately 20 min. We anticipate that the effects of minor vibrations during bag unfolding will subside within 1 h.

297298

303

## 4. Field deployment

- We deployed the developed branch enclosure system in two field settings to evaluate its performance. The first
- involved multiple Japanese cedar individuals from different regional populations growing in a common garden.
- The second consisted of a diurnal measurement on a mature individual tree.

#### 4.1 Measurements in a common garden

- The enclosure system was tested on Japanese cedar trees grown in a common garden setup at the Kawatabi Field
- Centre, Tohoku University, using cedar trees grown from cuttings of natural populations collected across Japan
- (Tsumura, 2022). Geographical differentiation in several functional traits of Japanese cedar has been revealed
- through common garden experiments (Hiura, 2022). We used cuttings derived from three populations: 9
- individuals from Ajigasawa (40.67° N, 140.20° E, 297m a.s.l.), 9 from Azoji (34.48° N, 131.96° E, 1060m a.s.l.),
- and 6 from Yakushima (30.33° N, 130.46° E, 1267m a.s.l.). Details of each population are provided in Kimura et
- al., (2014). The mean annual temperature at Kawatabi is 10.5 °C and the mean annual precipitation is 1697 mm.
- Field measurements were conducted on live, intact branches, with one south-facing branch selected per tree at
- approximately 1.3 m above the ground. Sampling was performed during daytime (9:00–15:00) from 29 May to 9
- June 2023, using a consistent protocol for all individuals.
- Measurements at the Kawatabi Field Centre detected 14 compounds, principally α-pinene, sabinene, β-farnesene,
- and ent-kaurene (Fig. 4). While total emission rates did not differ significantly among populations (ANOVA, P =
- 0.417, F=0.913), we observed substantial inter-individual variation. Specifically, the total emission rates varied by
- several orders of magnitude among individuals. Such high variability in emission rates is a well-documented
- characteristic of *C. japonica* (Saito et al., 2022; Matsunaga et al., 2011; Miyama et al., 2019; Tani et al., 2024),
- supporting the biological origin of this variation.
- In addition to the total rates, there was also considerable variation in the emission compositions (Fig. 4b). For
- example, in some trees (AJ016, AJ017, AJ020, AJ025, AJ033, AZ018, AZ019, AZ040, YK025) MTs accounted
- for more than 50% of the emission composition. In contrast, other individuals (AJ002, AJ035, AZ002, AZ004,
- AZ006, AZ024, AZ025, AZ029, AZ036, YK005, YK007, YK013, YK032, YK070) showed profiles where SQTs
- and DTs accounted for more than 50% of the emission composition.

Figure 4 Field experiment results showing BVOC emissions from Japanese cedar (*Cryptomeria japonica*). (a) Total BVOC emission rates from different populations (Yakushima [YK], Azouji [AZ], Ajigasawa [AJ]). (b) Chemical composition of BVOCs measured from individual trees from three populations.

Among all trees,  $\beta$ -farnesene was the most dominant SQT detected. Concentrations of  $\beta$ -farnesene varied

significantly. Among trees emitted β-farnesene (all individuals with the exception of YK025, YK070), β-farnesene accounted for 2.6~100% of total emissions. β-Farnesene is a crucial volatile organic compound, exerting significant influence on interactions between plants, aphids, and predator insects (Wang et al., 2024). β-Farnesene emission can be triggered by parasitism (Kivimäenpää et al., 2020). This compound is also suggested to be involved in responses to abiotic stresses, and it may help prevent leaf necrosis caused by local ozone reduction (Palmer-Young et al., 2015). Therefore, we can infer that trees with high β-farnesene emission rates may be experiencing some form of stress. BVOC emissions are broadly influenced by various biotic and abiotic factors (Holopainen and Gershenzon, 2010; Loreto and Schnitzler, 2010). Therefore, the variations in overall emission characteristics observed here may be attributed to environmental differences in the field and responses to biotic stress. However, the main objective of this study was the development of methods, and the sample size is not large enough to draw any conclusions about factor determining BVOC emissions. Further sampling will be necessary to clarify these factors, while detailed analyses of the relationships with phyllosphere microbial communities are being conducted elsewhere (Ishizaki et al., in submission).

#### 4.2 Diurnal variation in BVOC emissions

To evaluate the diurnal variation of BVOC emissions and assess the system's response to environmental conditions, we conducted additional measurements on a Japanese cedar tree (height: approximately 10 m) growing on the premises of the National Institute for Environmental Studies, Japan. Based on preliminary observations indicating that the BVOC profile of this individual was dominated by monoterpenes (MTs), the measurements focused specifically on MT emissions. Sampling was conducted at multiple time points over a 24-hour period on 5 August 2025, from the lowermost branch of the tree.

Throughout the night and morning (01:00–11:00), the emission rate increased with rising air temperature (Fig. 5a). The relationship between temperature and emissions during this pre-noon period (Fig. 5b) exhibited a typical exponential response, consistent with established models (e.g., Guenther et al., 1993). The temperature response coefficient ( $\beta$ , °C<sup>-1</sup>) calculated from this data using Equation (2) was 0.191. It is worth noting that our  $\beta$  value was derived using non-linear regression, which differs from the log-transformed linear regression method used in some previous studies on this species (e.g., Matsunaga et al., 2011; Okumura et al., 2013). For comparison, applying the linear method to our data yields a  $\beta$  of 0.143, which agrees well with their reported values (0.09–0.17). This result demonstrates that our method can quantitatively and accurately assess the standard environmental responses of plants.

In the afternoon, however, as a heatwave caused the temperature to exceed 40°C, the emission rate surged dramatically, deviating from the morning trend (Fig. 5c). Notably, even after the temperature decreased to approximately 30°C in the evening, the emission rate did not return to the level predicted by the standard temperature dependency (Fig. 5b). This hysteresis suggests that the surge was not a simple thermal response but was likely triggered by factors such as heat-induced physiological damage, as reported by Nagalingam et al. (2024). Although a mechanistic investigation of this single case is outside the scope of this methodological paper, it highlights the system's capability to capture plant responses to extreme weather events. This interesting phenomenon warrants further investigation.

Figure 5. Diurnal variation of monoterpene emissions from a field-grown Japanese cedar (Cryptomeria japonica) tree.

(a) Time series of emission rates (E; stacked bars) for monoterpene compounds and air temperature in the enclosure bag (red line), with an inset highlighting the 01:00–11:00 period. (b, c) Scatter plots of the emission rate versus temperature for data (b) before 11:00 and (c) after 11:00, respectively. The color of the markers indicates the time of day (hour).

#### 5. Conclusion

We successfully developed a portable, field-deployable system for measuring BVOC emissions from conifers. By using scrubbed ambient air as a purge gas and implementing a foldable bag design, the system addresses the logistical and methodological challenges associated with conventional techniques. The system maintained stable environmental conditions, provided BVOC- and ozone-free air, and ensured high measurement repeatability, making it suitable for field applications.

Field testing revealed significant variability in BVOC emissions both among individual trees and across

| 385 | geographical origins. These findings highlight the complex interplay of multiple factors in determining BVOC        |
|-----|---------------------------------------------------------------------------------------------------------------------|
| 386 | emission characteristics. This variability emphasizes the critical need for individual-level assessments to enhance |
| 387 | our understanding of BVOC dynamics in diverse ecosystems.                                                           |
| 388 | By providing a practical and reliable tool for BVOC measurement under natural conditions, the system can advance    |
| 389 | our understanding of the ecological roles and atmospheric effects of BVOCs, supporting studies of their             |
| 390 | spatiotemporal patterns and climatic interactions.                                                                  |
| 391 |                                                                                                                     |
| 392 | Data availability                                                                                                   |
| 393 | Data are available at 10.5281/zenodo.14965367, or upon request by contacting the corresponding authors.             |
| 394 | Supplement                                                                                                          |
| 395 | The supplement related to this article is available on-line at:                                                     |
| 396 | Author contributions                                                                                                |
| 397 | SJA and TS developed and maintained the instruments. YS and YT designed and managed the common gardens.             |
| 398 | YO and TS collected and analysed the samples. TIK provided assistance with data analysis. TS and TH supervised      |
| 399 | the research. YO analysed the data and wrote the manuscript with contributions from all coauthors.                  |
| 400 | Competing interests                                                                                                 |
| 401 | The authors declare that they have no conflict of interest.                                                         |
| 402 | Disclaimer                                                                                                          |
| 403 | Acknowledgements                                                                                                    |
| 404 | We would like to thank the staff of the Kawatabi Field Science Centre, Tohoku University, and the Mountain          |
| 405 | Science Centre, Tsukuba University, for their invaluable assistance and support during the fieldwork. We also       |
| 406 | sincerely appreciate H. Kenmoku and T. Ashitani for providing the chemical standards.                               |
| 407 | Financial support                                                                                                   |
| 408 | This work was supported by JSPS KAKENHI Grant Numbers JP21H02227, JP24K01809, JP21H05316 (to TH),                   |
| 409 | JP23H04965, and JP23H04969 (to TS).                                                                                 |
| 410 | References                                                                                                          |
| 411 | Arimura, G. Ichiro, Muroi, A., and Nishihara, M.: Plant-plant communications, mediated by (E)-β-ocimene             |
| 412 | emitted from transgenic tobacco plants, prime indirect defense responses of lima beans, Journal of Plant            |

Interactions, 7(3), 193-196, https://doi.org/10.1080/17429145.2011.650714, 2012.

Birami, B., Bamberger, I., Ghirardo, A., Grote, R., Arneth, A., Gaona-Colmán, E., Nadal-Sala, D., and Ruehr, N.

Aleppo pine, Oecologia, 197(4), 939–956, https://doi.org/10.1007/s00442-021-04905-y, 2021.

K.: Heatwave frequency and seedling death alter stress-specific emissions of volatile organic compounds in

15

- Bourtsoukidis, E., Pozzer, A., Williams, J., Makowski, D., Peñuelas, J., Matthaios, V. N., Lazoglou, G., Yañez-
- Serrano, A. M., Lelieveld, J., Ciais, P., Vrekoussis, M., Daskalakis, N., and Sciare, J.: High temperature
- sensitivity of monoterpene emissions from global vegetation, Commun. Earth Environ., 5, 23,
- https://doi.org/10.1038/s43247-023-01175-9, 2024.
- Bourtsoukidis, E., Guenther, A., Wang, H., Economou, T., Lazoglou, G., Christodoulou, A., Christoudias, T.,
- Nölscher, A., Yañez-Serrano, A. M., and Peñuelas, J.: Environmental Change Is Reshaping the Temperature
- Sensitivity of Sesquiterpene Emissions and Their Atmospheric Impacts, Glob. Change Biol., 31, e70258,
- https://doi.org/10.1111/gcb.70258, 2025.
- Bouvier-Brown, N. C., Holzinger, R., Palitzsch, K., and Goldstein, A. H.: Large emissions of sesquiterpenes and
- methyl chavicol quantified from branch enclosure measurements, *Atmospheric Environment*, 43(2), 389–401,
- https://doi.org/10.1016/j.atmosenv.2008.08.039, 2009.
- Chatani, S., Matsunaga, S. N., and Nakatsuka, S.: Estimate of biogenic VOC emissions in Japan and their effects
- on photochemical formation of ambient ozone and secondary organic aerosol, Atmospheric Environment, 120,
- 38–50, https://doi.org/10.1016/j.atmosenv.2015.08.086, 2015.
- Chatani, S., Okumura, M., Shimadera, H., Yamaji, K., Kitayama, K., and Matsunaga, S. N.: Effects of a detailed
- vegetation database on simulated meteorological fields, biogenic VOC emissions, and ambient pollutant
- concentrations over Japan, Atmosphere, 9(5), 179, https://doi.org/10.3390/atmos9050179, 2018.
- Dada, L., Stolzenburg, D., Simon, M., Fischer, L., Heinritzi, M., Wang, M., Xiao, M., Vogel, A. L., Ahonen, L.,
- Amorim, A., Baalbaki, R., Baccarini, A., Baltensperger, U., Bianchi, F., Daellenbach, K. R., Devivo, J., Dias,
- A., Dommen, J., Duplissy, J., Finkenzeller, H., Hansel, A., He, X.-C., Hofbauer, V., Hoyle, C. R.,
- Kangasluoma, J., Kim, C., Kürten, A., Kvashnin, A., Mauldin, R., Makhmutov, V., Marten, R., Mentler, B.,
- Nie, W., Petäjä, T., Quéléver, L. L. J., Saathoff, H., Tauber, C., Tome, A., Molteni, U., Volkamer, R., Wagner,
- R., Wagner, A. C., Wimmer, D., Winkler, P. M., Yan, C., Zha, Q., Rissanen, M., Gordon, H., Curtius, J.,
- Worsnop, D. R., Lehtipalo, K., Donahue, N. M., Kirkby, J., Haddad, I. E., and Kulmala, M.: Role of
- sesquiterpenes in biogenic new particle formation, Sci Adv., 2023 Sep 8;9(36), eadi5297,
- https://doi.org/10.1126/sciadv.adi5297, 2023.
- Ghirardo, A., Xie, J., Zheng, X., Wang, Y., Grote, R., Block, K., Wildt, J., Mentel, T., Kiendler-Scharr, A., Hallquist,
- M., Butterbach-Bahl, K., and Schnitzler, J. P.: Urban stress-induced biogenic VOC emissions and SOA-
- forming potentials in Beijing, Atmospheric Chemistry and Physics, 16(5), 2901–2920,
- https://doi.org/10.5194/acp-16-2901-2016, 2016.
- Guenther, A. B., Zimmerman, P. R., Harley, P. C., and Monson, R. K.: Isoprene and monoterpene emission rate
- variability' model evaluations and sensitivity analyses, Journal of Geophysical Research, 98(D7), 12609-
- 12617, https://doi.org/10.1029/93JD00527, 1993.
- Guenther, A. B., Jiang, X., Heald, C. L., Sakulyanontvittaya, T., Duhl, T., Emmons, L. K., and Wang, X.: The
- model of emissions of gases and aerosols from nature version 2.1 (MEGAN2.1): An extended and updated
- framework for modeling biogenic emissions, Geoscientific Model Development, 5(6), 1471–1492,
- https://doi.org/10.5194/gmd-5-1471-2012, 2012.
- Helin, A., Hakola, H., and Hellén, H.: Optimisation of a thermal desorption-gas chromatography-mass
- spectrometry method for the analysis of monoterpenes, sesquiterpenes and diterpenes, Atmospheric
- *Measurement Techniques*, 13(7), 3543–3560, https://doi.org/10.5194/amt-13-3543-2020, 2020.

- Helmig, D., Daly, R. W., Milford, J., and Guenther, A.: Seasonal trends of biogenic terpene emissions, *Chemosphere*, 93(1), 35–46, https://doi.org/10.1016/j.chemosphere.2013.04.058, 2013.
- Hiura, T.: Functional biogeography in Japanese cedar, *Ecological Research*, 38(1), 42-48, https://doi.org/10.1111/1440-1703.12321, 2022.
- Hiura, T., Yoshioka, H., Matsunaga, S. N., Saito, T., Kohyama, T. I., Kusumoto, N., Uchiyama, K., Suyama, Y.,
- and Tsumura, Y.: Diversification of terpenoid emissions proposes a geographic structure based on climate
- and pathogen composition in Japanese cedar, *Scientific Reports*, 11(1), 8307, <a href="https://doi.org/10.1038/s41598-">https://doi.org/10.1038/s41598-</a>
- 021-87810-X, 2021.
- Holopainen, J. K., and Gershenzon, J.: Multiple stress factors and the emission of plant VOCs, *Trends in Plant Science*, 15(3), 176–184, https://doi.org/10.1016/j.tplants.2010.01.006, 2010.
- Japan Metrological Agency: https://www.jma.go.jp/jma/menu/menureport.html, last access: 20 January 2025.
- Jardine, Kolby J., Raquel F. Zorzanelli, Bruno O. Gimenez, Luani Rosa de Oliveira Piva, Andrea Teixeira, Clarissa
- G. Fontes, Emily Robles, Niro Higuchi, Jeffrey Q. Chambers, and Scot T. Martin.: Leaf Isoprene and
- Monoterpene Emission Distribution across Hyperdominant Tree Genera in the Amazon Basin,
- Phytochemistry, 175 (July), 112366, https://doi.org/10.1016/j.phytochem.2020.112366, 2020.
- Kimura, M. K., Uchiyama, K., Nakao, K., Moriguchi, Y., Jose-Maldia, L. S., and Tsumura, Y.: Evidence for cryptic northern refugia in the last glacial period in Cryptomeria japonica, Annals of Botany, 114(8), 1687–1700,
- https://doi.org/10.1093/aob/mcu197, 2014.
- Kivimäenpää, M., Babalola, A. B., Joutsensaari, J., and Holopainen, J. K.: Methyl salicylate and sesquiterpene
- emissions are indicative for aphid infestation on Scots pine, Forests, 11(5), 573,
- <u>https://doi.org/10.3390/F11050573</u>, 2020.
- Laothawornkitkul, J., Taylor, J. E., Paul, N. D., and Hewitt, C. N.: Biogenic volatile organic compounds in the Earth system, *New Phytologist*, 183(1), 27–51, https://doi.org/10.1111/j.1469-8137.2009.02859.X, 2009.
- Li, L., Guenther, A. B., Xie, S., Gu, D., Seco, R., Nagalingam, S., and Yan, D.: Evaluation of semi-static enclosure
- technique for rapid surveys of biogenic volatile organic compounds (BVOCs) emission measurements,
- *Atmospheric Environment*, 212, 1–5, https://doi.org/10.1016/j.atmosenv.2019.05.029, 2019.
- Loreto, F., and Schnitzler, J. P.: Abiotic stresses and induced BVOCs, *Trends in Plant Science*, 15(3), 154–166, https://doi.org/10.1016/j.tplants.2009.12.006, 2010.
- Matsunaga, S. N., Mochizuki, T., Ohno, T., Endo, Y., Kusumoto, D., and Tani, A.: Monoterpene and sesquiterpene
- emissions from Sugi (Cryptomeria japonica) based on a branch enclosure measurements, Atmospheric
- *Pollution Research*, 2(1), 16–23, https://doi.org/10.5094/APR.2011.003, 2011.
- Matsunaga, S. N., Chatani, S., Nakatsuka, S., Kusumoto, D., Kubota, K., Utsumi, Y., Enoki, T., Tani, A., and Hiura,
- T.: Determination and potential importance of diterpene (kaur-16-ene) emitted from dominant coniferous
- trees in Japan, *Chemosphere*, 87(8), 886–893, https://doi.org/10.1016/j.chemosphere.2012.01.040, 2012.
- Matsunaga, S. N., Niwa, S., Mochizuki, T., Tani, A., Kusumoto, D., Utsumi, Y., Enoki, T., and Hiura, T.: Seasonal
- variation in basal emission rates and composition of mono- and sesquiterpenes emitted from dominant
- conifers in Japan, Atmospheric Environment, 69, 124–130. https://doi.org/10.1016/j.atmosenv.2012.12.004,
- 2013
- Mochizuki, T., Endo, Y., Matsunaga, S., Chang, J., Ge, Y., Huang, C., and Tani, A.: Factors affecting monoterpene

- emission from Chamaecyparis obtusa, Geochem. J., 45, E15–E22, <a href="https://doi.org/10.2343/geochemj.1.0130">https://doi.org/10.2343/geochemj.1.0130</a>, 498 2011.
- Miyama, T., Tobita, H., Uchiyama, K., Yazaki, K., Ueno, S., Saito, T., Matsumoto, A., Kitao, M., and Izuta, T.:

  Differences in monoterpene emission characteristics after ozone exposure between three clones representing
  major gene pools of Cryptomeria japonica, J. Agric. Meteorol., 74, 102–108,
  https://doi.org/10.2480/agrmet.D-17-00043, 2018.
- Miyama, T., Tobita, H., Uchiyama, K., Yazaki, K., Ueno, S., Uemura, A., Matsumoto, A., Kitao, M., and Izuta, T.:
   Seasonal Changes in Interclone Variation Following Ozone Exposure on Three Major Gene Pools: An
   Analysis of Cryptomeria Japonica Clones, Atmosphere, 10(11), 643, https://doi.org/10.3390/atmos10110643,
   2019.
- Monson, R. K., Trahan, N., Rosenstiel, T. N., Veres, P., Moore, D., Wilkinson, M., Norby, R. J., Volder, A., Tjoelker,
  M. G., Briske, D. D., Karnosky, D. F., and Fall, R.: Isoprene emission from terrestrial ecosystems in response
  to global change: minding the gap between models and observations, Philos. Trans. R. Soc. Math. Phys. Eng.
  Sci., 365, 1677–1695, https://doi.org/10.1098/rsta.2007.2038, 2007.
- Nagalingam, S., Wang, H., Kim, S., and Guenther, A.: Unexpectedly strong heat stress induction of monoterpene, methylbutenol, and other volatile emissions for conifers in the cypress family (Cupressaceae), Science of The Total Environment, 956, 177336, https://doi.org/10.1016/j.scitotenv.2024.177336, 2024.
- Namdari, M., Lee, C. S., and Haghighat, F.: Active ozone removal technologies for a safe indoor environment: A comprehensive review, *Building and Environment*, 187, 107370, https://doi.org/10.1016/j.buildenv.2020.107370, 2021.
- Núñez, L., Plaza, J., Pérez-Pastor, R., Pujadas, M., Gimeno, B. S., Bermejo, V., and García-Alonso, S.: High water vapour pressure deficit influence on Quercus ilex and Pinus pinea field monoterpene emission in the central Iberian Peninsula (Spain), Atmos. Environ., 36, 4441–4452, https://doi.org/10.1016/s1352-2310(02)00415-6, 2002.
- Ogasa, M. Y., Utsumi, Y., Miki, N. H., Yazaki, K., and Fukuda, K.: Cutting stems before relaxing xylem tension induces artefacts in Vitis coignetiae, as evidenced by magnetic resonance imaging, Plant Cell Environ., 39, 329–337, https://doi.org/10.1111/pce.12617, 2016.
- Okumura M., Ise T., Tani A., Miyama T., Kominami Y., Tohno S.: Effect of leaf temperature and light intensity on monoterpene emissions from japanese cedar (Cryptomeria japonica), Eco-Engineering, 25(4), 117-121, 2013.
- Ortega, J., and Helmig, D.: Approaches for quantifying reactive and low-volatility biogenic organic compound emissions by vegetation enclosure techniques - Part A, *Chemosphere*, 72(3), 343-364, https://doi.org/10.1016/j.chemosphere.2007.11.020, 2008.
- Ortega, J., Helmig, D., Daly, R. W., Tanner, D. M., Guenther, A. B., and Herrick, J. D.: Approaches for quantifying reactive and low-volatility biogenic organic compound emissions by vegetation enclosure techniques Part B: Applications, *Chemosphere*, 72(3), 365–380, https://doi.org/10.1016/j.chemosphere.2008.02.054, 2008.
- Palmer-Young, E. C., Veit, D., Gershenzon, J., and Schuman, M. C.: The sesquiterpenes(E)-β-farnesene and (E)α-bergamotene quench ozone but fail to protect the wild tobacco Nicotiana attenuata from ozone, UVB, and drought stresses, *PLoS ONE*, 10(6), https://doi.org/10.1371/journal.pone.0127296, 2015.
- Peñuelas, J., and Staudt, M.: BVOCs and global change, *Trends in Plant Science*, 15(3), 133–144, https://doi.org/10.1016/j.tplants.2009.12.005, 2010.

- Praplan, A. P., Tykkä, T., Schallhart, S., Tarvainen, V., Bäck, J., and Hellén, H.: OH reactivity from the emissions of different tree species: Investigating the missing reactivity in a boreal forest, *Biogeosciences*, *17*(18), 4681–4705, https://doi.org/10.5194/bg-17-4681-2020, 2020.
- Randall, D., Artaxo, P., Bretherton, C., Feingold, G., Forster, P., Kerminen, V., Kondo, Y., Liao, H., Lohmann, U.,
- Rasch, P., Satheesh, S., Sherwood, S., Stevens, B., Zhang, X., Qin, D., Plattner, G., Tignor, M., Allen, S.,
- Boschung, J., Nauels, A., Xia, Y., Bex, V., Midgley, P., Boucher, O., and Randall, D.: Clouds and Aerosols.
- In: Climate Change 2013: The Physical Science Basis. Contribution of Working Group I to the Fifth
- Assessment Report of the Intergovernmental Panel on Climate Change Coordinating Lead Authors: Lead
- Authors, 2013.
- Rotstayn, L. D., Keywood, M. D., Forgan, B. W., Gabric, A. J., Galbally, I. E., Gras, J. L., Luhar, A. K., McTainsh,
- G. H., Mitchell, R. M., and Young, S. A.: Possible impacts of anthropogenic and natural aerosols on
- Australian climate: A review, International Journal of Climatology, 29(4), 461-479,
- https://doi.org/10.1002/joc.1729, 2009.
- Saito, T., Kusumoto, N., and Hiura, T.: Relation of leaf terpene contents to terpene emission profiles in Japanese
- $cedar \textit{(Cryptomeria japonica), Ecological Research, 38(1), 74-82, \underline{https://doi.org/10.1111/1440-1703.12323},$
- 2022.
- Saunier, A., Mpamah, P., Biasi, C., and Blande, J. D.: Microorganisms in the phylloplane modulate the BVOC
- emissions of Brassica nigra leaves, *Plant Signaling and Behavior*, 15(3), 1728468.
- https://doi.org/10.1080/15592324.2020.1728468, 2020.
- Shirota, T.: Studies on the plasticity of tree crowns based on an analysis of branch age distribution in Japanese cedar, pp. 194, Dissertation (Kyusyu University)(in Japanese), 2000.
- Stoy, P. C., Trowbridge, A. M., Siqueira, M. B., Freire, L. S., Phillips, R. P., Jacobs, L., Wiesner, S., Monson, R.
- K., and Novick, K. A.: Vapor pressure deficit helps explain biogenic volatile organic compound fluxes from
- the forest floor and canopy of a temperate deciduous forest, Oecologia, 197, 971–988,
- https://doi.org/10.1007/s00442-021-04891-1, 2021.
- Tani, A., Masui, N., Chang, T. W., Okumura, M., and Kokubu, Y.: Basal emission rates of isoprene and
- monoterpenes from major tree species in Japan: interspecies and intraspecies variabilities, *Progress in Earth*
- and Planetary Science, 11(1), 42, https://doi.org/10.1186/s40645-024-00645-8, 2024.
- Tsumura, Y.: Genetic structure and local adaptation in natural forests of Cryptomeria japonica, *Ecological Research*, 64-73, https://doi.org/10.1111/1440-1703.12320, 2022.
- Umebayashi, T., Ogasa, M. Y., Miki, N. H., Utsumi, Y., Haishi, T., and Fukuda, K.: Freezing xylem conduits with
- liquid nitrogen creates artifactual embolisms in water-stressed broadleaf trees, Trees, 30, 305–316,
- https://doi.org/10.1007/s00468-015-1302-4, 2016.
- Yañez-Serrano, A., Penuelas, J., Jorba, O., Graeffe, F., Meder, M., Garmash, O., Zhang, Y., Li, H., Luo, Y., Praplan,
- A., Hellen, H., Schobesberger, S., Vettikkat, L., Thomas, S., Kurtén, T., Taipale, D., Bourtsoukidis, E.,
- Guenther, A., and Ehn, M.: Unaccounted impacts of diterpene emissions on atmospheric aerosol loadings,
- https://doi.org/10.21203/rs.3.rs-5407662/v1, Research Square [preprint], 13 November 2024.
- Wang, B., Jacquin-Joly, E., and Wang, G.: The role of (e)-β-farnesene in tritrophic interactions: biosynthesis,
- chemoreception, and evolution, *Annual Review of Entomology*, 70, 313-335, https://doi.org/10.1146/annurev-
- ento-013024-021018, 2024.