# Peer review of "Field-deployable branch enclosure system for biogenic volatile organic compounds emitted from conifers"

_EGUsphere, 2025_

## Author Comment (AC1)

We sincerely thank Dr. Praplan for the careful review and encouraging comments. Our detailed responses are provided below, with the *reviewer's comments* and our replies distinguished by formatting. The line numbers referenced in our responses correspond to those in the revised manuscript.

*Comment1:- lines 211-212: 'After trimming its base, we cut the branch under water to maintain it vascular integrity.' Could the author explain a little bit more how the branch was cut under water and possibly provide a reference demonstrating how vascular integrity is maintained by doing so?*

Response1: We agree that our explanation regarding the procedure of cutting branches under water was insufficient. We have added following sentences in the main manuscript.

L232: After trimming the base, the branch was re-cut under water to maintain its vascular integrity, and the cut end was kept submerged throughout the experiment. This underwater cutting technique is a standard method to prevent air from the xylem vessels, which can cause cavitation and disrupt water transport (Ogasa et al., 2016; Umebayashi et al., 2016). Indeed, measurements on detached branches represent a well-established approach in BVOC research (e.g., Jardine et al., 2020), including for coniferous species with large storage pools similar to C. japonica (Mochizuki et al., 2011; Miyama et al., 2018). Furthermore, Monson et al. (2007) demonstrated that this method maintains stable rates of photosynthesis, stomatal conductance, and isoprene emission for detached branches, showing no significant differences from branches that remained attached to the tree. As *C. japonica* emits stored rather than de novo synthesized BVOCs, and the distance from the cut site to the enclosed section of the branch was sufficiently long (60 cm), the effect of cutting on our measurements is considered negligible.

*Comment2: - lines 226-227: 'at least one terpene was detected in each category'. Why did the author decided not to included all the detected terpenes (one in each category) in Fig. 2? There seems to be only MTs and one DT.*

Response2: Thank you for this comment, which has highlighted an ambiguity in our manuscript. First, Figure 2 does present all of the terpenes that were detected in the experiment. We recognize that our wording was confusing. Our original intention was to state that both the thermal desorption and solvent extraction methods successfully detected compounds, thus validating the use of both techniques. However, we agree that this sentence was not essential and could be misinterpreted. To improve the clarity of the

manuscript, we have removed this sentence entirely.

*Comment3:- Figure 4: I am not sure to understand the boxplot (panel (a)) as there are datapoints scattered horizontally (why?) and some blue dots are on the same levels as gray crosses. It is not clear from the caption if the crosses are outliers, but if they are, why are there blue dots (not outliers?) at the same height? In panel (b), the three colors used for MTs are very similar and make it difficult to see what compounds are present in the emissions from the figure.*

Response3: We agree that the presentation was confusing, and we have revised both panels accordingly.

For Figure 4 (a): Our original figure superimposed a jittered scatter plot onto the boxplot, which caused the confusing horizontal distribution and resulted in outliers appearing twice (once as a boxplot outlier, once as a scatter point). We acknowledge this was misleading. We have revised the figure to a standard boxplot format, showing only the outliers as individual points, which makes the plot much clearer. Additionally, we changed the y-axis units to $\mu g\ (gdw)^{-1}\ h^{-1}$ to avoid the large numbers (e.g., 50,000) of the original ng-based scale and improve readability.

For Figure 4 (b): We also agree that the colors for the monoterpenes were too similar. We have addressed this by selecting a new, more distinct color palette in the revised figure to ensure each compound can be easily distinguished.

The revised version is provided below for reference.

[Figure]

[Figure]

*Comment4: - lines 296-303: The authors mention the possible effect of stress, but state that it is not the objective of their study to look closer at the factors determining BVOCs emissions. The sample size, they argue, is 'not large enough', but I believe that it is still a decent enough sample size as they have shown using various statistical tools. As a suggestion (more than a request for revision), I think that it would be nice to include something about the environmental conditions (e.g. temperature and its effect on the emission rates) as the sensors (for temperature, radiation, etc.) are part of the dynamic branch enclosure system and it would be good to demonstrate what conclusions could be made with the acquired dataset. I understand, however, if the authors have planned to*

*demonstrate this in a subsequent manuscript with a larger dataset and more solid conclusions.*

Response4: Thank you for your constructive comments. We agree that exploring the system's ability to capture environmental responses is a crucial aspect of its validation.
In our main field campaign (Chapter 4.1), the primary objective was to assess inter-individual variation. For this reason, we normalized all emission rates to a standard temperature to minimize temperature-induced variability and better resolve the underlying biological differences between trees.
However, we also recognize the importance of demonstrating the system's capability to track environmental drivers, a point also raised by Reviewer #2. Therefore, in response to the reviews, we conducted an additional field experiment specifically designed to monitor the diurnal variation of BVOC emissions from a single, intact tree.
These new results have been added as a new section (4.2) and figure (Fig. 5). This new section provides a clear demonstration of what can be concluded from our dataset regarding environmental responses, directly addressing your suggestion. While a more detailed investigation with a larger dataset is part of our future plans, we believe this addition significantly strengthens the manuscript by validating the system's performance under dynamic, field conditions. We appreciate your encouragement.

*Comment5: In addition to my previous comments, I would like to add that, for the dataset published, it would be good to have for the BVOC data the inclusion as metadata of what units apply the numbers that are reported.*

Response5: Thank you for your comments. As per your comment, we have added units to the published dataset.

---

## Author Comment (AC2)

We sincerely thank the reviewer for the careful review and constructive comments. Our detailed responses are provided below, with the *reviewer's comments* and our replies distinguished by formatting. The line numbers referenced in our responses correspond to those in the revised manuscript.

**General Comments**

*Comment1:* ***System validation and applicability.*** *The central claim of the manuscript is that the system enables portable measurements from multiple trees within a single day. However, most of the performance evaluations (e.g., reproducibility and stabilization time) are based on measurements from cut branches, which do not represent intact physiological conditions. Cut branches are known to alter emission profiles, especially in species with large internal storage pools such as conifers. Given that this system is meant to overcome such limitations, a convincing demonstration under field conditions using live, rooted trees is essential. Otherwise, this system would not differ significantly from simple, well-stablished chamber-based measurements in laboratory. The authors should also clarify whether the system is designed to be reused across trees or if multiple enclosure collars need to be installed in advance. Discussing a field-based example of multi-tree sampling in practice would help substantiate this important advantage.*

Response1: We sincerely thank Reviewer #2 for this insightful and crucial feedback. We agree that demonstrating the system's performance on live, intact trees in the field is essential to substantiate our central claim of portability and multi-tree sampling.

1. Why we used cut branches for method validations

We wish to clarify our rationale for using excised branches for some of the initial validation tests. To precisely evaluate the system's intrinsic performance (e.g., reproducibility), stable emission rates were required, which is very difficult to achieve in the field due to fluctuating environmental conditions. We did consider using potted saplings in a controlled environment, but found them unsuitable because their BVOC emissions were too low. Consequently, we opted for excised branches from mature trees, as they provided the most realistic and stable conditions for these specific validation purposes.

Our approach aligns with a large body of literature. Measurements on detached branches are a well-established method in BVOC research for short-term experiments where emission rates are not significantly affected by excision (e.g. Jardine et al., 2020), especially for coniferous species with large internal storage pools like Cryptomeria

japonica (e.g., Mochizuki et al., 2011; Miyama et al., 2018; Jardine et al., 2020). To minimize any potential artifacts from this widely used method, we followed standard best-practice protocols. As detailed in the manuscript (L232), we used long branches and performed underwater re-cutting to maintain vascular integrity (Monson et al., 2007). We believe this approach ensured that our initial validation was both robust and reliable.

2. Additional measurements on live, rooted tree

Second, we conducted additional field measurements on live, rooted trees. These new results (section 4.2), which we detail in response to your next comment, validate our system's performance under field conditions.

3. Clarification on field operation

Third, regarding field usability, the collars are indeed reusable. They are installed the day before measurement and can be moved between trees. In this study, five measurement sets were used during the field campaign. As such, this system allows for the sampling of multiple individuals in settings where electricity is not available. To clarify, the following sentences were added:

L114: The system is designed for efficiently sampling multiple trees in a single day. This is achieved by pre-installing the support collars on each target tree the day before, and then moving the main portable enclosure apparatus between these collars on the sampling day. In this study, five sets of collars were used to sample five trees. The detailed procedure for enclosing a single branch is as follows:

*Comment2: **Ambiguity in field deployment.** The field deployment data show emission rates spanning up to six orders of magnitude among individuals of the same species. While biological variability is expected, such a wide range raises questions about system consistency. The authors attribute this variability to individual differences, but without clearer evidence that the technique itself is not contributing to it (e.g. via consistently different handling of the samples from the three areas - perhaps this is why the same tree species are so consistently different among the different locations), this interpretation remains uncertain. If the final field deployment data were also based on detached branches, then the system has not yet been demonstrated under its intended real-world conditions, and the results would not validate the system's field applicability as claimed.*

Response2: We thank the reviewers for raising this important point and for the opportunity to clarify our field methodology and results.

1. **Clarification on field measurement conditions**

    First, we would like to clarify that all field measurements in section 4 were conducted from intact branches of live, rooted trees, not from excised branches, as shown in the photo in Figure 1(b).

2. **Evidence of system stability and consistent methodology**

    To demonstrate the stability of our system, we have added new data from repeated measurements conducted on the same branch (see new Figure S1 in the Supplement). These results show high reproducibility, which strongly indicates that our measurement system is stable and reliable, and that the system itself is not the source of the large variability observed between different trees.

    Furthermore, all measurements were conducted in a common garden. In this setting, trees from different provenances grow under identical environmental conditions, and were sampled during the same period. The sampling protocol and handling were applied consistently across all individuals. This experimental design makes it highly unlikely that the observed differences are artifacts of location or inconsistent methodology.

    We have revised the sentences in 3.3. measurement repeatability:

    L258: In addition to the tests on excised branches, we conducted a reproducibility experiment on an intact branch of a live seedling. The same branch was measured twice, and the results are compared in Figure S1. The measurements showed reasonable consistency for all detected compounds, supporting the stability of our system. We note, however, that this test was conducted on a young seedling with a limited number of emitted compounds.

3. **Consistency with known biological variability**

    The high degree of variability we observed is well-documented characteristic of Japanese ceder. Several previous studies have reported similarly large, order-of-magnitude differences in BVOC emission rates among individuals of this species (Saito et al., 2022; Matsunaga et al., 2011; Miyama et al., 2019; Tani et al.,2024). Therefore, our results are consistent with the known biological variability for this species.

In summary, based on (a) our standardized experimental design that minimized methodological artifacts, (b) the demonstrated stability of our system from repeated measurements, and (c) the consistency of our results with the known high biological variability of the species, we attribute the observed range of emission rates to the

inherent genetic and physiological differences among the individual trees.
To make clear the above points, we have revised the sentences in 4. Field deployment:

L311: Field measurements were conducted on live, intact branches, with one south-facing branch selected per tree at approximately 1.3 m above the ground. Sampling was performed during daytime (9:00–15:00) from 29 May to 9 June 2023, using a consistent protocol for all individuals.

L314: Measurements at the Kawatabi Field Centre detected 14 compounds, principally α-pinene, sabinene, β-farnesene, and ent-kaurene (Fig. 4). While total emission rates did not differ significantly among populations (ANOVA, P = 0.417, F=0.913), we observed substantial inter-individual variation. Specifically, the total emission rates varied by several orders of magnitude among individuals. Such high variability in emission rates is a well-documented characteristic of C. japonica (Saito et al., 2022; Matsunaga et al., 2011; Miyama et al., 2019; Tani et al., 2024), supporting the biological origin of this variation.
In addition to the total rates, there was also considerable variation in the emission compositions (Fig. 4b). For example, in some trees (AJ016, AJ017, AJ020, AJ025, AJ033, AZ018, AZ019, AZ040, YK025) MTs accounted for more than 50% of the emission composition. In contrast, other individuals (AJ002, AJ035, AZ002, AZ004, AZ006, AZ024, AZ025, AZ029, AZ036, YK005, YK007, YK013, YK032, YK070) showed profiles where SQTs and DTs accounted for more than 50% of the emission composition.

*Comment3: **Lack of environmental response validation.** A core requirement for validating a new BVOC enclosure system is demonstrating that it can reproduce known patterns such as diurnal variations and emission responses to temperature and light. The manuscript does not include any environmental-driven validation. Without observing characteristic temporal emission patterns (e.g. the temperature and light-driven increases during day), it is difficult to distinguish between physiological emissions and stress-induced pulses caused by handling or storage depletion. At least a clear diurnal cycle from a rooted field-grown tree is required for validating such new measurement technique.*

Response3: Thank you for your constructive comments. In direct response to your

feedback, we conducted an additional measurement specifically designed to monitor the diurnal variation of BVOC emissions from a live, rooted tree. We have added these new results to the manuscript as a new section (chapter 4.2) and figure (Fig. 5).

The new measurements revealed two distinct phases. During the pre-noon period, emissions systematically tracked the rise in temperature, showing a clear temperature dependency, and the calculated temperature coefficient (β) is consistent with the established literature for this species. This demonstrates the system's consistency and its ability to monitor a standard physiological process within a single individual. It provides strong evidence that the large variability observed in the common garden study was indeed due to inter-individual differences, not system instability.

In contrast, the afternoon was characterized by a non-linear emission surge during a heatwave, a response that deviated from the initial temperature dependency. This highlights the system's capability to also capture stress-related environmental responses to extreme events.

This new experiment complements our initial work in the common garden (Fig. 4), where the primary objective was to compare intraspecific variation across a consistent midday period (09:00–15:00), rather than to characterize diurnal cycles.

We are confident that the addition of this dedicated diurnal study, which confirms the system's ability to measure meaningful biological responses, significantly strengthens the manuscript and fully addresses your concern.

L346:

4.2 Diurnal variation in BVOC emissions

To evaluate the diurnal variation of BVOC emissions and assess the system's response to environmental conditions, we conducted additional measurements on a Japanese cedar tree (height: approximately 10 m) growing on the premises of the National Institute for Environmental Studies, Japan. Based on preliminary observations indicating that the BVOC profile of this individual was dominated by monoterpenes (MTs), the measurements focused specifically on MT emissions. Sampling was conducted at multiple time points over a 24-hour period on 5 August 2025, from the lowermost branch of the tree.

Throughout the night and morning (01:00–11:00), the emission rate increased with rising air temperature (Fig. 5a). The relationship between temperature and emissions during this pre-noon period (Fig. 5b) exhibited a typical exponential response, consistent with established models (e.g., Guenther et al., 1993). The temperature response coefficient (β, °C$^{-1}$) calculated from this data using Equation (2) was 0.191. It is worth noting that

our β value was derived using non-linear regression, which differs from the log-transformed linear regression method used in some previous studies on this species (e.g., Matsunaga et al., 2011; Okumura et al., 2013). For comparison, applying the linear method to our data yields a β of 0.143, which agrees well with their reported values (0.09–0.17). This result demonstrates that our method can quantitatively and accurately assess the standard environmental responses of plants.

In the afternoon, however, as a heatwave caused the temperature to exceed 40°C, the emission rate surged dramatically, deviating from the morning trend (Fig. 5c). Notably, even after the temperature decreased to approximately 30°C in the evening, the emission rate did not return to the level predicted by the standard temperature dependency (Fig. 5b). This hysteresis suggests that the surge was not a simple thermal response but was likely triggered by factors such as heat-induced physiological damage, as reported by Nagalingam et al. (2024). Although a mechanistic investigation of this single case is outside the scope of this methodological paper, it highlights the system's capability to capture plant responses to extreme weather events. This interesting phenomenon warrants further investigation.

**Specific Comments**

*Comment4: **L24-26**. Please note that observing significant individual variation cannot not demonstrate system reliability (quite the opposite actually). This should be reworded to avoid conflating biological variation with instrument performance.*

Response4: Thank you for this crucial point. We completely agree that our original wording conflated biological variation with system reliability, which was a logical error. We have reworded this sentence in the abstract to correct this and to more accurately describe our findings.
The revised sentence now reads:

L24: Field testing with Japanese cedar (Cryptomeria japonica) demonstrated the system's robust field performance, successfully capturing both significant inter-individual variability and the dynamic diurnal patterns of BVOC emissions. The system's ability to reliably resolve these differences under field conditions demonstrates its applicability for advancing our understanding of BVOC dynamics in diverse ecosystems.

*Comment5: **L37-42.** Please consider expanding this paragraph and referencing recent review articles covering emission behaviour of monoterpenes (eg. https://doi.org/10.1038/s43247-023-01175-9) , sesquiterpenes (e.g. https://doi.org/10.1111/gcb.70258), and diterpenes (e.g. https://doi.org/10.21203/rs.3.rs-5407662/v1), to provide more context on their chemical properties and relevance.*

Response5: Thank you very much for introducing interesting references. We have added the suggested review articles in the paragraph.

L48: Recent comprehensive reviews have further underscored the critical and distinct roles of these terpene classes in biosphere-atmosphere interactions (Bourtsoukidis et al., 2024, 2025; Yañez-Serrano et al., 2024). For instance, diterpenes are now understood to be particularly potent contributors to SOA formation, potentially having a disproportionately large impact relative to their emission rates (Yañez-Serrano et al., 2024). Moreover, these reviews highlight that the emission rates and composition of MTs and SQTs can vary significantly among individuals, which may reflect diverse adaptive strategies to environmental stresses (Bourtsoukidis et al., 2024, 2025). To untangle the complex factors governing these emissions, a dual approach of broad-scale analysis and detailed, individual-level data collection is essential.

*Comment6: **L154-163. A.** The emission rate equation differs from more commonly used formulations (e.g., E = F × (Cout − Cin)/(dry weight mass)). Please elaborate on the reasoning behind this approach and its comparability.*

Response6: Thank you for your constructive comments. Our original formulation was non-standard and not sufficiently clear. To improve clarity, we have revised the manuscript to use the standard mass-balance equation as follows:

L166: The rate of BVOC emission (E, in ng $(gdw)^{-1}$ $h^{-1}$) was first calculated using a mass-balance equation:

$$E = \frac{F \times (C_{\text{out}} - C_{\text{in}})}{W_{\text{dry}}} \tag{1}$$

where F is the flow rate of purge air through the enclosure (L $h^{-1}$); Cout is the BVOC concentration in the air exiting the enclosure (ng $L^{-1}$), determined from the mass of the

compound collected on a sorbent tube divided by the total volume of air sampled; Cin is concentration of BVOCs in the incoming purge air, determined from a blank measurement (an empty enclosure); Wdry is the dry weight of the enclosed branch (g dw), estimated from Shirota (2000).

To allow for comparison across measurements taken at different temperatures, this measured emission rate (E) was then normalized to a basal emission rate (Es, in ng $(gdw)^{-1} h^{-1}$) at a standard temperature (Ts, 30°C = 303.15 K), following the algorithm of Guenther et al. (1993):

$$E_s = \frac{E}{\exp[\beta(T - T_s)]} \tag{2}$$

where T is the temperature inside the enclosure, and β is an empirical coefficient that quantifies the temperature sensitivity of emissions. The β values used were 0.17 for MTs, 0.20 for SQTs, and 0.21 for DTs (Matsunaga et al., 2011, 2012, 2013).

*Comment7:* **B.** *The use of basal emission rate (ES) calculated using fixed β values from the literature assumes consistent temperature sensitivity across all conditions. This approach is not appropriate in a study designed to evaluate natural emissions. Empirical derivation of temperature responses would provide more convincing validation.*

Response7: Thank you for this valid point. We agree that empirically deriving the β value from our own data would be the most robust approach.
However, our field campaign was conducted over a short period of 11 consecutive days, resulting in a limited temperature range. The coefficient of variation for the measured temperatures was only 0.15 (RSD), indicating low temperature variability.
With such small temperature range and limited size of our dataset, it was not suitable for a robust empirical derivation of β. Therefore, we used established β values from the literature as the more reasonable and necessary step for normalization in this context.
We have revised the manuscript to clearly state this limitation and provide the context of the narrow temperature range.

L179: It should be noted that these β values were not empirically derived from our own dataset, as the measurements were conducted over a narrow temperature range that was unsuitable for robust parameterization.

*Comment8: **L195-196.** The statement that the system reduces "desiccation stress" based on chamber humidity is incorrect. Drought stress is primarily soil-driven, and relative humidity in the enclosure does not replicate root water availability. Please rephrase or remove this statement.*

Response8: Thank you for this accurate and important correction. Our use of "drought stress" was incorrect. We have removed this term from the manuscript. Our intended meaning was that using ambient air, which is more humid than dry cylinder air, reduces the atmospheric water demand on the branch. To describe this phenomenon accurately, we have revised the text to refer to vapor pressure deficit, a key driver of BVOC emissions.

L212: The humidity within the enclosure more closely reflected the external humidity when purge air was supplied via our air delivery system compared to using dry cylinder gas (Table 2). This reduces the atmospheric water demand on the branch, avoiding potential artifacts caused by an artificially high vapor pressure deficit.

*Comment9: **L204/Table 2.** With the flows used, one would have expected higher humidity inside the chamber as the result of evapotranspiration from a living branch.*

Response9: Thank you for your comments.  The moderate humidity levels observed were an expected result of our experimental design.
We used a photographic white umbrella to diffuse and reduce the intensity of direct sunlight on the enclosure. This, combined with our system's efficient air exchange (residence time of approx. 1.6 minutes), prevents both overheating and the accumulation of transpired water vapor, which explains the stable conditions observed.

L104: A photographic white umbrella was used to diffuse and reduce the intensity of direct sunlight on the enclosure during purging and sampling.
L220: This overall stability in both temperature and humidity is attributed to two key design features. First, the photographic white umbrella was used to diffuse and reduce the intensity of direct sunlight, preventing overheating of the enclosed branch. Second, the high air exchange rate, with a residence time of approximately 1.6 minutes, effectively prevents the accumulation of both heat and transpired water vapor.

*Comment10: **L210-216.**   The orders of magnitude of phyllocladane stronger emissions is perhaps an indication that we are seeing the effects of stress and not natural emissions.*

Response10: You are correct that such a large emission warrants careful consideration. While we acknowledge that stress can influence BVOC emissions, we believe the primary reason for these particularly high phyllocladane emissions is biological rather than a measurement artifact. Our interpretation is based on previous findings (Saito et al., 2022) which show that significant phyllocladane emission from *C. japonica* is a trait specific to individuals that have this compound stored in their tissues. The high emissions were observed only in certain individuals, which is consistent with this understanding.

As this paragraph focuses specifically on the reproducibility of the measurement system, we feel that a detailed discussion of the biological interpretation of specific compound emissions is beyond its scope. However, we appreciate you bringing this to our attention.

Comment11: **Chapter 3.4 / Figure 3.** *The sharp emission peak followed by exponential decay likely reflects the depletion of storage pools in a severed branch rather than natural stabilization. Comparing late-stage emissions to initial peaks does not validate reproducibility but rather shows a system with low emission rates as the storage pools are emptying. Demonstrating that with isoprene, which is mainly de novo produced, would have been more convincing.*

Response11: Thank you for your comments. First, regarding the use of isoprene, this was not possible as our study species, Cryptomeria japonica, does not emit de novo synthesized BVOCs such as isoprene and relies exclusively on storage pools. Second, we argue that the observed decay is not due to storage pool depletion. The total amount of terpenes emitted during the measurement (3.75 µg/gdw) accounts for only 0.024% of the estimated total storage (approx. 15.9 mg/gdw, based on Saito et al., 2022). This fraction is too small to cause depletion. Instead, we interpret the pattern as a transient emissions pulse caused by mechanical disturbances of installing the chamber, a phenomenon reported by Mochizuki (2011). Therefore, the subsequent stabilization demonstrates the system's ability to measure a steady baseline emission rate once this initial disturbance subsides.

Comment12: **Chapter 4.** *As mentioned above, please clarify whether the field deployment involved measurements from branches still attached to living trees. This is a key point for assessing whether the system has been tested in realistic conditions.*

Response12: Thank you for seeking this clarification. We confirm that all field deployment measurements were conducted on intact branches of living trees. We have now made this point more explicit in the manuscript to avoid any ambiguity.

L311: Field measurements were conducted on live, intact branches, with one south-facing branch selected per tree at approximately 1.3 m above the ground.